# High-Molecular-Weight Exopolysaccharides Production from *Tuber brochii* Cultivated by Submerged Fermentation

**DOI:** 10.3390/ijms24054875

**Published:** 2023-03-02

**Authors:** Cheng-Chun Chen, Parushi Nargotra, Chia-Hung Kuo, Yung-Chuan Liu

**Affiliations:** 1Department of Chemical Engineering, National Chung Hsing University, Taichung 402, Taiwan; 2Department of Seafood Science, National Kaohsiung University of Science and Technology, Kaohsiung 811, Taiwan; 3Center for Aquatic Products Inspection Service, National Kaohsiung University of Science and Technology, Kaohsiung 811, Taiwan

**Keywords:** *Tuber borchii*, submerged fermentation, exopolysaccharide, gel permeation chromatography, Fourier-transform infrared spectroscopy, β-(1–3)-glucan

## Abstract

Truffles are known worldwide for their peculiar taste, aroma, and nutritious properties, which increase their economic value. However, due to the challenges associated with the natural cultivation of truffles, including cost and time, submerged fermentation has turned out to be a potential alternative. Therefore, in the current study, the cultivation of *Tuber borchii* in submerged fermentation was executed to enhance the production of mycelial biomass, exopolysaccharides (EPSs), and intracellular polysaccharides (IPSs). The mycelial growth and EPS and IPS production was greatly impacted by the choice and concentration of the screened carbon and nitrogen sources. The results showed that sucrose (80 g/L) and yeast extract (20 g/L) yielded maximum mycelial biomass (5.38 ± 0.01 g/L), EPS (0.70 ± 0.02 g/L), and IPS (1.76 ± 0.01 g/L). The time course analysis of truffle growth revealed that the highest growth and EPS and IPS production was observed on the 28th day of the submerged fermentation. Molecular weight analysis performed by the gel permeation chromatography method revealed a high proportion of high-molecular-weight EPS when 20 g/L yeast extract was used as media and the NaOH extraction step was carried out. Moreover, structural analysis of the EPS using Fourier-transform infrared spectroscopy (FTIR) confirmed that the EPS was β-(1–3)-glucan, which is known for its biomedical properties, including anti-cancer and anti-microbial activities. To the best of our knowledge, this study represents the first FTIR analysis for the structural characterization of β-(1–3)-glucan (EPS) produced from *Tuber borchii* grown in submerged fermentation.

## 1. Introduction

Truffles are ectomycorrhizal fungi that belong to the phylum Ascomycota, the order Pezizales, and the family Tuberaceae genus *Tuber* [1,2]. Truffles live in symbiotic relationships with the roots of certain trees, where they coexist and produce hypogenous fruiting bodies. True truffles are those that belong to the genus *Tuber*, while other fruiting bodies, such as from the genus *Melanogaster*, are called false truffles, truffle-like, etc. [2,3]. Truffles are globally known as the most economically valuable edible fungi due to their distinctive taste, aroma, and nutritional characteristics [1,4]. They are a great source of vitamins, minerals, essential amino acids, polysaccharides, phenolic compounds, essential fatty acids, sterols, sphingolipids, and androstanolone. Additionally, they are also an untapped source of medicinal compounds with anti-inflammatory, antioxidant, antibacterial, immune-suppressant, anti-mutagenic, anti-carcinogenic, antiviral, and hepato-protective properties. The presence of medicinally important molecules and biomedical properties shows their promising potential for application in pharmacology and medicine [5,6,7,8]. Moreover, they, like other fungi [9,10,11,12], also produce various enzymes, such as cellulase, xylanase, amylase, laccase, and so on, during their growth cycle to breakdown/degrade the complex organic matter present around the host plant into simple compounds [5].

Like many other species of truffles, *Tuber borchii* is highly appreciated for its flavor and sensorial qualities [1]. It is a whitish truffle that can be found throughout Europe thriving in both sub-alkaline and acidic soils in climates ranging from cold temperate to Mediterranean. The host trees and shrubs for its growth include oak, poplar, strawberry, and pine, among others [4]. Its ecological attributes and the gastronomic properties of its fruiting bodies contribute to its significance. It is regarded as the *Tuber* species with the largest ecological magnitude due to its ability to adapt to various environmental conditions. Therefore, it can be planted in an environment that is unfavorable for other species of truffles, thereby increasing the geographical area where truffles can be cultivated [13]. Moreover, unlike other *Tuber* species, its mycelium can easily by isolated and cultivated in vitro [14], which makes it a desired *Tuber* species on which to execute various biochemical analyses for in-depth study. *T. borchii* is also known for its ability to produce aroma due to the presence of various volatile organic compounds, which can be exploited for commercial production of truffle-flavored food with natural aroma [15]. Since *T. borchii* has not been used for the production of mycelial biomass, exopolysaccharides (EPSs), and intracellular polysaccharides (IPSs) in submerged fermentation, and due to its aforementioned properties, it was chosen in the current study.

The majority of truffles are obtained by either natural or semi-artificial cultivation. The natural cultivation is impacted by various environmental conditions, such as the temperature, soil type, and host tree. [16]. Additionally, harvesting the truffle fruiting body is a time-consuming and laborious process, with inconsistency in truffle quality, and typically takes 4 to 12 years [4]. Therefore, to address the growing global market demand and reduce the overexploitation of natural truffles, which is endangering the species, in-depth research is being carried out on the semi-artificial cultivation of truffle fruiting bodies. In this regard, submerged fermentation has emerged as a viable alternative for the effective production of bioactive mycelia. Submerged fermentation can help overcome the limitations of natural cultivation by offering a shorter production time and higher biomass amount [17]. Moreover, it results in the production of various EPSs, IPSs, enzymes, and other metabolites of commercial importance [4]. Extracellular amylase, cellulase, and xylanase production from *Tuber maculatum* mycelia has been reported by submerged fermentation with varying culture conditions [1,5,18].

The polysaccharides produced by truffles are known to have different biomedical properties, including immunomodulating, anti-tumor, hypoglycemic, and blood-lipid-lowering activities, and do not affect normal cells [8]. Truffles contain either α- or β-glucan, or mixed α-, β-glucan, which mainly includes (1→3)/(1→6)-β-D-glucans, (1→3)-α-D-glucans, and (1→6)-β-D-glucan. The β-D-glucans are reported to increase the production of macrophages, natural killer cells, T cells, and cytokines, and a few possess antioxidant activity. Moreover, they are also regarded as the utmost significant immunomodulating polysaccharides. Heteroglycans, which consist of sugars, such as mannose, fructose, galactose, and glucose, are also present in the truffle [19]. The molecular weight (MW) of polysaccharides has a critical impact on their biological function. The biological activity of polysaccharides with a high MW may generally be superior. The enhanced activity of high-MW polysaccharides is attributed to their effective binding affinity towards the carbohydrate receptors of immune cells. Polysaccharides with low MW have the potential to penetrate into immune cells and generate an immune response [20]. However, the antioxidant property of a polysaccharide is inversely associated with its MW, i.e., polysaccharides with lower molecular exhibit greater antioxidant activity, whereas high-MW polysaccharides are thought to have potent immunomodulatory and anti-tumor activities [20,21,22].

Therefore, considering the importance of polysaccharides production from truffles and the role of MW in determining the biomedical properties of a polysaccharide, *T. borchii* was cultivated in submerged fermentation in the current study. The effects of different medium compositions (carbon and nitrogen sources) and different culture conditions on the growth of truffle mycelia and polysaccharides production was evaluated to find the optimal growth conditions for enhanced truffle mycelia and polysaccharides production. The MW distribution and the structure of EPS were estimated by using Gel Permeation Chromatography (GPC) and Fourier-transform infrared spectroscopy (FTIR), respectively.

## 2. Results and Discussion

### 2.1. Effect of Different Carbon Sources on Mycelial Biomass and Polysaccharides Production

Living organisms utilize different carbon sources for energy, growth, and development. Therefore, in this study, four different sugars, i.e., two monosaccharides (glucose and fructose) and two disaccharides (sucrose and lactose), were screened to determine the best carbon source. It was found that the dry cell weight (DCW, 3.47 g/L) of truffle with sucrose as the carbon source was significantly higher than the other three carbon sources after 28 days of submerged fermentation (Figure 1). The DCW obtained in the presence of glucose, fructose, and lactose was 2.52 g/L, 2.14 g/L, and 0.67 g/L, respectively (Figure 1). The poor mycelial growth observed in the presence of lactose could possibly be due either to the inability of β-galactose (one of the monomers of lactose) to favor mycelial growth or to low β-galactosidase activity expression in truffle *T. brochii* to hydrolyze lactose into glucose and galactose before it can be used effectively as a carbon source [23]. A study reported galactose, along with lactose and maltose, to be unsuitable for the growth of a medicinal mushroom of genus *Phellinus*, as they had less impact on promoting mycelial growth [24].

In terms of polysaccharides production, the presence of glucose, fructose, sucrose, and lactose resulted in the EPS amount of 0.192 ± 12 g/L, 0.178 ± 7 g/L, 0.233 ± 1.78 g/L, and 0.165 ± 9 g/L, respectively, after 28 days of submerged fermentation. The consumption of sucrose as a carbon source resulted in a maximum EPS production that was about 1.41 times higher than the lowest amount produced with lactose as a media component (0.165 ± 9 g/L). A similar pattern of results was obtained in the case of IPS production. The maximum IPS production was obtained in the presence of sucrose (0.838 ± 32.3 g/L), followed by glucose (0.735 ± 27.1 g/L) and fructose (0.619 ± 99.5 g/L). The presence of lactose in the medium resulted in the lowest IPS production (0.316 ± 6.2 g/L), which was about 2.648 times lower than that obtained from sucrose (Figure 1). The results indicated that the production of polysaccharides was proportional to the total amount of mycelia. Since the presence of sucrose resulted in maximum growth and EPS production, it was chosen as the carbon source for further studies. Research studies are available for the use of various carbon sources to produce mycelial biomass, EPS, and IPS from different mushrooms. A study reported the use of different carbon sources (glucose, sucrose, lactose, and fructose) to assess the impact of these sources on the mycelial dry weight and EPS and IPS production from *Pleurotus* sp. mushroom grown in submerged fermentation [25]. It was observed that the maximum mycelium dry weight was observed in the presence of glucose; however, the highest amount of EPS and IPS was obtained in the presence of sucrose [25]. In another study, although sucrose’s presence as the carbon source resulted in increased biomass and IPS production from *Tuber sinense* in submerged fermentation, the maximum EPS was produced in the presence of lactose, even with poor cell growth [23]. The result of the present study correlates with those reported in the literature indicating that the choice and consumption of carbon sources vary among different mushroom varieties for mycelial growth and polysaccharides production.

When various carbon sources were used in the submerged cultivation, the substrate yields for both biomass and EPS were also estimated, which are listed in Table 1. As shown in the table, sucrose gave the highest yields for both biomass and EPS, whereas lactose displayed a very low biomass yield. The EPS yield for lactose, however, remained at a specific level of 0.41 (%), resulting in the high biomass-based EPS yield of 0.245 g/g. It is assumed that the lactose is not fit for *T. brochii* growth due to the aforementioned reasons, whereas the slow growth of cells did not appear to prevent the release of EPS from cells. This special high yield of EPS based on biomass could provide us a basis for further exploration when using lactose as a carbon source in *Tuber* cultivation.

### 2.2. Effect of Sucrose Concentration

The impact of varying sucrose concentrations (40 g/L, 50 g/L, 60 g/L, 80 g/L, and 100 g/L) on the amount of mycelial biomass and polysaccharides production was also investigated. The amount of DCW analyzed after 28 days of growth was found to be 3.47 g/L, 3.25 g/L, 3.37 g/L, 3.58 g/L, and 3.31 g/L at the concentration (g/L) of 40, 50, 60, 80, and 100, respectively (Figure 2). Although there was not much difference in the mycelium amount, however, the maximum mycelial growth was observed at the sucrose concentration of 80 g/L. With regard to EPS production, the maximum amount (0.547 ± 28.34 g/L) was obtained at the concentration of 80 g/L and decreased to 0.464 ± 31.51 g/L at the 100 g/L sucrose concentration (Figure 2). The EPS obtained at a 50 g/L and 60 g/L concentration was 0.386 ± 46.95 g/L and 0.429 ± 39.54 g/L, respectively. The EPS amount at 80 g/L sucrose was 2.196 times higher compared to the lowest (0.249 ± 4.14 g/L), which was obtained at 40 g/L.

The amount of IPS at the sucrose concentrations (g/L) of 50, 60, and 100 were 0.987 ± 64.29 g/L, 1.071 ± 24.82 g/L, and 0.884 ± 29.82 g/L, respectively (Figure 2). Moreover, at the sucrose concentration of 80 g/L, the IPS production was up to 1.142 ± 36.16 g/L, which was 1.304 times higher than the lowest amount obtained at 40 g/L (0.875 ± 5.1 g/L). The results indicated that the polysaccharides production is significantly affected by the sucrose concentration. The decrease in mycelial biomass and polysaccharides production at 100 g/L sucrose could be attributed to the increased viscosity of the fermentation medium due to the increased amount of the carbon source, which may limit the nutritional and oxygen diffusion, represses mycelial growth, and consequently lead to low polysaccharides production. Moreover, the osmotic pressure brought on by the high sugar content may reduce polysaccharides production [18,23]. The results were in line with the reported study on polysaccharides production from *Grifola umbellate* in submerged fermentation. The cell density and polysaccharides production was enhanced, with an increased glucose concentration from 1% to 3%; however, at a further increase from 3% to 5% concentration, the cell density and polysaccharides production was inhibited [26]. Accordingly, sucrose at the concentration of 80 g/L was used as a carbon source in subsequent experiments.

### 2.3. Effect of Different Nitrogen Sources on Mycelial Biomass and Polysaccharides Production

Several different organic and inorganic nitrogen sources were screened for their effects on mycelial biomass and polysaccharides production. A lack of nitrogen hinders the growth of fungi and the production of polysaccharides, since it is a crucial component of proteins and nucleic acids. The organic sources of nitrogen used were yeast extract, peptone, and malt extract, while ammonium chloride (NH_4_Cl) and diammonium phosphate ((NH_4_)_2_HPO_4_) were inorganic nitrogen sources. The nitrogen sources were used at a concentration of 10 g/L, and sucrose (80 g/L) was used as a carbon source. After 28 days of submerged fermentation, the presence of yeast extract as a nitrogen source in the medium resulted in a DCW of 3.41 ± 0.04 g/L. Compared with yeast extract, the DCW decreased in the presence of peptone (1.40 ± 0.05 g/L), followed by malt extract (0.41 ± 0.01 g/L) and NH_4_Cl (0.27 ± 0.04 g/L). However, when (NH_4_)_2_HPO_4_ was used, a much lower final cell density (DCW), i.e., 0.08 ± 0.01 g/L, was obtained (Figure 3).

The presence of yeast extract in the medium also favored EPS production, as the maximum EPS of 0.498 ± 13.24 g/L was achieved, which was about 3.45 times higher than the lowest amount obtained in the presence of (NH_4_)_2_HPO_4_ (0.144± 16.15 g/L). EPS of 0.382 ± 23.98 g/L, 0.333 ± 11.38 g/L, and 0.179 ± 11.59 g/L was obtained when peptone, malt extract, and NH_4_Cl were used as a nitrogen source, respectively. In terms of IPS production, a similar trend was found, with the highest amount of IPS (0.929 ± 12.36 g/L) in the presence of yeast extract as compared to peptone (0.286 ± 40.31 g/L), malt extract (0.051 ± 5.55 g/L), and NH_4_Cl (0.036 ± 18.5 g/L). When (NH_4_)_2_HPO_4_ was used as a nitrogen source, no IPS was detected due to the negligible mycelial growth. From these findings it was evident that the organic nitrogen may have promoted cell growth and polysaccharide formation by supporting the synthesis of essential amino acids since the cell density and polysaccharides production were low in the presence of inorganic nitrogen. Moreover, yeast extract is known to be a rich source of proteins, amino acids, and vitamins, which favors cell growth [27,28]. Contrary to our results, some reports suggest that different nitrogen sources may stimulate high mycelial growth and polysaccharides production from the same organisms under the same conditions. Wang and co-workers [28] observed that the maximum mycelial biomass (2.32 g/L), EPS (1.58 g/L), and IPS (29.1 mg/L) were obtained when yeast extract, malt extract, and peptone, respectively, were used as a nitrogen source for the cultivation of *Grifola frondosa* under submerged fermentation. Similarly, the mycelium dry weight (27.3 ± 2.1 g/L) was maximum when yeast extract was present in the medium, whereas meat extract produced the maximum EPS (24.1 ± 0.8 mg/mL) and IPS (5.23 ± 0.07 mg/mL) amount when *Pleurotus* sp. was grown in submerged fermentation using these nitrogen sources [25].

### 2.4. Effect of Yeast Extract Concentration

Since yeast extract promoted mycelia growth and polysaccharides production, it was used at varying concentrations (g/L), i.e., 10, 15, 20, 25, and 30, to further identify its optimal concentration for enhanced mycelial growth and polysaccharides production. The results indicated that cell growth increased with an increase in yeast extract concentration and was highest at 20 g/L with a DCW of 4.06 ± 0.11 g/L (Figure 4). A decrease in growth was observed upon further increasing the concentration, with the lowest DCW at 30 g/L yeast extract (1.27 ± 0.22 g/L). In contrast to mycelial growth, EPS production increased with the increasing concentration and was found to be maximum (0.905 ± 4.85 g/L) when 30 g/L yeast extract was used in the fermentation medium. However, as seen in Figure 4 and Appendix A, for IPS production, the concentration of 20 g/L yeast extract gave a maximum IPS of 1.385 ± 41.03 g/L, and on further increasing the concentration, a negative effect on IPS production was observed. IPS production at 20 g/L yeast extract was 3.37 times greater than the lowest amount obtained at 30 g/L (0.411 ± 103.05 g/L). The high amount of IPS at 20 g/L could be attributed to the high mycelial growth obtained at the same concentration. Liu et al. [23] reported that the maximum dry cell weight (19.53 ± 0.13 g/L), EPS production titer (0.66 ± 0.06 g/L), and IPS production titer (0.90 ± 0.03 g/L) was obtained at a 25 g/L concentration of yeast extract from *Tuber sinense* in submerged fermentation. In another study, yeast extract as the organic nitrogen source and ammonium chloride as the inorganic nitrogen source resulted in the highest mycelium dry weight and EPS production from *Lentinus squarrosulus* [29]. From the findings of the current study, it was evident that the maximum mycelial biomass and IPS production was obtained at the 20 g/L yeast extract concentration; however, EPS production was highest with 30 g/L yeast extract. Therefore, in the subsequent experiments, the analysis of the molecular weight of the EPS produced with different yeast extract concentrations using the GPC technique was considered as the basis to ascertain the optimum yeast extract concentration.

### 2.5. Effect of Carbon/Nitrogen Ratio

The concentration of both carbon and nitrogen sources and their ratio in the medium is highly crucial for enhanced cell growth and polysaccharides (EPS and IPS) production. Therefore, the carbon/nitrogen (C/N) ratios using sucrose and yeast extract as the carbon and nitrogen sources, respectively, were determined. The effect of different C/N ratios on the mycelial growth and EPS and IPS production is shown in Table 2. When the C/N ratio was higher than 4, as in case I, the DCW obtained was almost 3.3–3.6 g/L, which was associated with higher IPS of about 0.880–1.150 g/L. In this case, the EPS level obtained was in the range of 0.250–0.550 g/L. However, in case II, when the C/N ratio was lower than 4, a lower DCW (1.3–1.7 g/L) was obtained, which was associated with a lower IPS amount (0.400–0.700 g/L). Nevertheless, in this case, higher EPS (0.770–0.900 g/L) was achieved. This indicated that whereas the production of IPS may be positively correlated with the quantity of cell biomass, EPS production appears to be adversely correlated with cell biomass. When the C/N ratio and sucrose concentration were 4 and 80 g/L (case II), respectively, it was possible to achieve both high cell biomass (4.06 ± 0.11 g/L) and a greater IPS level (1.385 ± 41.03 g/L). Additionally, in this case, the EPS obtained was 0.725 ± 30.17 g/L. This suggested that not only the C/N ratios, but also the carbon source levels are critical for both biomass and polysaccharides (EPS and IPS) production. Similar to our study, the C/N ratio was important for the biomass and polysaccharides production from mushroom *Pleurotus* sp. A C/N ratio of 80:1 yielded enhanced mycelial biomass and IPS and EPS production of 20.1 g/L, 11.89 ± 1.15 mg/mL, and 3.38 ± 1.21 mg/mL, respectively [25].

### 2.6. Truffle Growth and Biomass/Polysaccharides Production

Generally, in a batch culture, the classical growth curve of all the fungi, including truffles, include different phases, i.e., lag phase, log phase, and stationary phase. These phases are different from one another on the basis of the amount of cell biomass, metabolite production, and nutrient depletion. The lag phase is characterized by the fact that not all of the cells from the inoculum begin to proliferate at the same time, followed by the log phase or exponential phase, wherein cells utilize nutrients and grow at an exponential rate. The stationary phase is represented by the exhaustion of nutrients and no further increase in the biomass amount, as cell growth slows down, and the number of living cells drops off exponentially. In order to identify the optimal time of incubation to achieve maximum growth and polysaccharides production, it is crucial to understand and monitor the time profile of the growth. Therefore, truffle was grown in submerged fermentation using the selected carbon and nitrogen sources in the optimized concentration, and the growth and polysaccharides production (EPS and IPS) was monitored for 35 days. Figure 5a shows the growth curve of truffle *T. borchii*, where the initial stage was from the 0th to 7th day, when the inoculum was in the adaptation period, and little growth was observed. Exponential growth was seen from the 11th to 28th day, as on the 28th day, the maximum mycelia biomass (5.38 ± 0.01 g/L), EPS (0.70 ± 0.02 g/L), and IPS (1.76 ± 0.01 g/L) were obtained (Figure 5a,b). A decline in the biomass growth and polysaccharides production was observed after the 28th day of submerged fermentation. Therefore, in the subsequent experiments, submerged fermentation was carried out for 28 days to obtain the maximum biomass and polysaccharides production.

### 2.7. Molecular Weight Determination of Exopolysaccharides

The molecular weight of a polysaccharide is an important parameter for its biological activity. It is generally believed that higher-MW polysaccharides typically exhibit improved biological and anti-cancer activity owing to the ability of polysaccharides with MWs > 90 kDa to form a triple-helix structure. Furthermore, polysaccharides with MW > 200 kDa are reported to be good immunogens with immunomodulating activity [30,31]. Therefore, in the present study, the MW of the EPS produced by the truffle after 28 days of fermentation was analyzed using the GPC method, using pullulan as the standard. The molecular weight of EPS produced at varying yeast extract concentrations on different days of submerged fermentation and extracted by various extraction methods, including water extraction, alkali extraction, and acid extraction, was estimated. The MW distribution of the polysaccharides was divided into high molecular weight (280 kDa > MW > 100 kDa), medium molecular weight (100 kDa > MW > 10 kDa), and low molecular weight (10 kDa > MW > 800 Da) within the calibration curve.

#### 2.7.1. Molecular Weight of EPS Produced at Different Concentrations of Yeast Extract

The results from the previous experiments showed that the maximum mycelial biomass was obtained at 20 g/L yeast extract; however, the best EPS amount was obtained at the highest concentration of yeast extract (30 g/L). Since the molecular weight of the polysaccharide also affects its functional activity, the molecular weight of EPS produced at different yeast extract concentrations was analyzed. The MW and molecular weight distribution of EPS extracted from water, NaOH, and HCl extraction were compared. It was found that the EPS extracted by using water, NaOH, and HCl at the yeast concentrations 15 g/L and 20 g/L had the maximum average MW. Figure 6a shows that the EPS from water extract was mostly distributed in the small-MW range and slightly in the large-molecular-weight range. The NaOH extraction of EPS at a 20 g/L yeast extract concentration contained EPS of higher and medium MW (Figure 6b). More than 70% of the EPS was in the small MW range when the concentration of yeast extract was 25 g/L. However, when HCl was used for the extraction, 90% of the EPS was in the small MW range, with an average MW of only about 5 kDa (Figure 6c). It was evident from the water and NaOH extraction of EPS that at a yeast concentration lower than 15 g/L and higher than 20 g/L, small-MW EPS was more prominent; therefore, only 2 yeast extract concentrations, i.e., 15 g/L and 20 g/L, were used for HCl extraction. The high ratio of small-molecular-weight EPS even at 15 and 20 g/L yeast extract could be attributed to the hydrolysis of EPS by HCL. From these findings it was concluded that a 20 g/L yeast concentration is the optimal concentration for obtaining high mycelial biomass and high-molecular-weight EPS.

#### 2.7.2. Molecular Weight of EPS Produced on Different Days of Submerged Fermentation

The maximum mycelial growth and polysaccharides production was achieved on the 28th day of submerged fermentation, after which both were decreased. However, since the polysaccharides production was different on different days of fermentation, there could be a possibility of variation in the MW of EPS produced during these days. Therefore, the relationship between EPS production and the MW of EPS produced on different days of fermentation was investigated. It was observed that the average MW of polysaccharides was gradually increased with fermentation time. On the 0th day of fermentation, the average MW of EPS was 47,349 Da, which increased to 92,445 Da on the 28th day, with the highest average MW achieved throughout the course of fermentation. The average MW remained stable to about 92 kDa, even after the 28th day of fermentation (Figure 7a). The EPS produced from day 0 to day 14 was mostly in the range of small MWs, with EPS of about 0.22 g/L on day 14. A shift from the small-MW range to the medium range was observed for EPS produced between the 18th to 21st day, and about 0.28 g/L EPS was recorded on the 21st day. However, the ratio of high-molecular-weight EPS increased after the 21st day and reached the average MW of 92 kDa on the 28th day of fermentation. A decrease in small- and medium-molecular-weight EPS was observed on the 28th day. The amount of high-MW EPS increased from about 0.053 g/L on day 0 to 0.32 g/L on the 28th day (Figure 7b). Similar to our study, Yeh et al. [32] also reported the increase in the ratio of high-MW EPS produced from *Hypocreales* sp. NCHU01 under submerged fermentation with the increase in the cultivation time. The average MW of EPS produced from edible fungus *Stropharia rugosoannulata* was reported to be 2.1 × 10^4^ D [33]. The high-MW EPS produced using the optimal concentration of the carbon and nitrogen source under submerged fermentation may possess multiple biomedical applications in the health care sector.

### 2.8. FTIR Polysaccharide Analysis

The FTIR spectroscopy technique was used to analyze the structural characterization of EPS produced by the truffle owing to its sensitivity to the position and anomeric arrangement of glycosidic bonds. Figure 8 represents the FTIR spectra of EPS produced by *T. borchii* at the sucrose and yeast extract concentrations of 80 and 20 g/L, respectively, extracted with NaOH and collected on the 28th day of cultivation by submerged fermentation. The broad and intense peak observed at 3372 cm^−1^ represented the stretching vibration of a hydroxyl group (O–H), suggesting the presence of a molecule with several free hydroxyl groups. The existence of the methylene group in the EPS was confirmed by the absorption peak at 2935 cm^−1^, which corresponded to the stretching vibration of C–H bonds [34]. The peak at 1648 cm^−1^ was assigned to the bound-water bending vibration, as the polysaccharides possess a significant affinity for water and are rapidly hydrated in the solid form [35]. The -CH_3_ stretching vibration was denoted by the presence of a peak at 1400 cm^−1^. The characteristic absorption peaks at 1114 and 881 cm^−1^ were attributed to (1→3) glycosidic linkages and a β-configuration in the anomeric region, which confirmed that the EPS was β-(1–3)-glucan [34,36]. The β-glucans are recognized to have antimicrobial, anti-cancer, and glucose-lowering properties. They are the strong immune response modifiers due to their ability to trigger innate and adaptive immune responses by interacting with a variety of immune cell receptors [6,36]. The production of β-glucans as EPS may pave the way for its biomedical application.

## 3. Materials and Methods

### 3.1. Strain and Chemicals

The truffle strain used in the present study was *Tuber borchii* (ATCC^®^ MYA1019™), which was purchased from the American Type Culture Collection (ATCC, Manassas, VA, USA). It was identified using molecular PCR (T100, Bio-rad, Hercules, CA, USA) analysis, and the strain identification report is added as Supplementary materials (Appendix A). The mycelium was maintained on a potato-dextrose-agar (PDA) plate. The PDA plate was inoculated with mycelium and incubated at 25 °C for 28 days for growth. The biomass was then grown in 500 mL Erlenmeyer flasks containing 100 mL of seed media with the following composition: 0.5% glucose, 0.1% yeast extract, 0.1% peptone, 0.02% KH_2_PO_4_, 0.02% MgSO_4_∙7H_2_O [26]. The seed media was inoculated with four pieces of cutter square (20 mm × 20 mm) mycelial biomass previously grown on a PDA plate. The cutter square was homogenized (V100, Osterizer, Mexico) prior to the seeding and selected from the outer part of the plate culture. The inoculated media was incubated on a rotary shaker set at 100 rpm and 25 °C for 21 days. Chemicals, including glucose, magnesium sulphate (MgSO_4_·7H_2_O), potassium dihydrogen phosphate (KH_2_PO_4_), and pullulan standard, were purchased from Showa Denko K.K, Tokyo, Japan. Bacto™ yeast extract and Vitamin B_1_ were acquired from Becton, Dickinson and Company, Sparks, MD, USA and Merck Ltd., Taipei, Taiwan, respectively. PDA was procured from HiMedia, Bombay, India, HCL and ethanol from Echo Chemical Co. Ltd., Miaoli, Taiwan, and NaOH from Showa, Saitama, Japan. All the chemicals used in the current study were of analytical grade.

### 3.2. Flask Cultivation

A shake-flask culture was performed in a 500 mL Erlenmeyer flask containing 100 mL of the basal fermentation medium. The medium consisted of the following components: glucose, 40 g/L; yeast extract, 10 g/L; KH_2_PO_4_, 1 g/L; MgSO_4_∙7H_2_O, 1 g/L; vitamin B_1_ hydrochloride, 0.15 g/L. The pH was adjusted to 7 by the addition of either 0.1 N HCl or 1 N NaOH. Media were sterilized at 121 °C for 20 min, and the carbon source was autoclaved separately. A total of 90 mL medium in a 500 mL shake flask was inoculated with 10 mL seed broth. The flasks were incubated on a rotary shaker at 100 rpm and 25 °C for 28 days.

### 3.3. Effect of Different Carbon, Nitrogen Sources, and Carbon/Nitrogen Ratio

Various carbon and nitrogen sources were screened for mycelial growth and polysaccharides production. The carbon sources used in the study were glucose, sucrose, fructose, and lactose. The effect of the carbon source was studied by using either of the carbon sources in basal fermentation medium at 40 g/L concentration and keeping other growth parameters and conditions constant. Similarly, the effect of various organic substrates (yeast extract, peptone, and malt extract) and inorganic salts (ammonium chloride (NH_4_Cl) and diammonium phosphate ((NH_4_)_2_HPO_4_)) as nitrogen sources was also evaluated by using either of the organic and inorganic nitrogen sources in the basal fermentation medium, as per the concentration mentioned above in Section 3.2. Furthermore, varying concentrations of the best carbon (40 g/L, 50 g/L, 60 g/L, 80 g/L, and 100 g/L) and nitrogen sources (10 g/L, 15 g/L, 20 g/L, 25 g/L, and 30 g/L), which resulted in the maximum growth and polysaccharides production (EPS and IPS), were used in order to find the optimal concentration of the respective source. Additionally, the effect of C/N ratio on mycelial biomass and polysaccharide (EPS and IPS) production was also evaluated. Two cases were studied. In case I, the carbon source concentration was varied, and the nitrogen source was used at a constant concentration with the following C/N ratios: 4 (40/10), 5 (50/10), 6 (60/10), 8 (80/10), and 10 (100/10). In case II, the concentration of the carbon source was maintained constant whereas the nitrogen source concentration was changed, resulting in the following ratios; 8 (80/10), 5.3 (80/15), 4 (80/20), 3.2 (80/25), and 2.66 (80/30).

### 3.4. Assays

#### 3.4.1. Cell Biomass

The cell biomass content was expressed as dry cell weight (DCW). Each time, three flasks were taken, and the broth was filtered to obtain the DCW. The filter paper used was Advantec No.1 with a pore size of 6 µm and a diameter of 90 mm (Toyo Roshi Kaisha Ltd., Tokyo, Japan). The cells were then washed three times with distilled water and dried at 60 °C to a constant weight.

#### 3.4.2. Exopolysaccharides Extraction and Analysis

The EPS amount was quantified by removing the mycelial biomass in the broth through filtration. To the filtered broth, 95% (*v*/*v*) ethanol (four times the volume of the broth) was added and kept at −20 °C overnight in order to precipitate out the crude EPS. The ethanol and broth mixture was then subjected to centrifugation (Megafuge 16R, Thermo Scientific, Taichung, Taiwan) at 10,000× *g* for 20 min. The supernatant was discarded and to the pellet, 75% ethanol (8 mL) was added, followed by centrifugation at 10,000× *g* for 20 min. The supernatant was removed, and the pellet was kept at 60 °C for 20 min to remove water and ethanol. For EPS extraction using water, HCl, or NaOH, the pellet was resuspended in deionized (DI) water, 1 N HCl, or 1 N NaOH, respectively, and kept at 85 °C and 100 rpm for 1 h. The total sugar concentration of the crude EPS was assayed using the phenol-sulfuric acid method, and glucose was used as a standard [37].

#### 3.4.3. Intracellular Polysaccharides Analysis

For the determination of IPS, the mycelial biomass obtained after filtration was subjected to freeze drying. A total of 100 mg of powder was mixed with 10 mL of distilled water, and the mixture was kept at 121 °C for 20 min. The process was repeated twice, followed by centrifugation at 8000 rpm for 10 min. The supernatant was collected and subjected to the same ethanol precipitation and washing process to obtain the crude IPS, as described in Section 3.4.2. The total sugar was estimated using the phenol-sulfuric acid method [37].

#### 3.4.4. Molecular Weight Analysis

The molecular weight of the crude EPS was determined on the basis of a calibration curve built by the pullulan standard (5.9–1600 kDa). For estimation, a gel permeation chromatography (Analytical Scientific Instruments, Richmond, CA, USA), together with a RI-2000 detector (Showa Denko K.K, Tokyo, Japan), was employed. Three columns, viz., KB-802.5, KB-804, and KB-805 (0.8 cm × 30 cm, Showa Denko K.K, Tokyo, Japan), were connected in a series. The flow rate was kept at 1.0 mL/min with the column temperature at 35 °C, and the injection volume was 20 mL. DI-water was utilized as a mobile phase [38]. The average molecular weight of the EPS extracted using the water extraction, alkali extraction, and acid extraction methods at different yeast extract concentrations was also analyzed. Additionally, the molecular weight of the polysaccharides produced at different fermentation time intervals was estimated to determine the average MW of the polysaccharides produced by truffle. The area percentage of the EPS with high MW was estimated using the following equation:$$\mathrm{Molecular}\;\mathrm{weight}\;\mathrm{area}\;\left(\%\right)=\frac{\mathrm{Area}\;\mathrm{of}\;\mathrm{EPS}\;\mathrm{with}\;\mathrm{MW}\;\mathrm{higher}\;\mathrm{than}\;300\;\mathrm{kDa}}{\mathrm{Area}\;\mathrm{of}\;\mathrm{total}\;\mathrm{EPS}}\times 100$$

The molecular weight distribution tendency of the EPS, along with the fermentation time, from small MW converted to medium MW and then to large MW was estimated by multiplying the polysaccharide amount (g/L) with the MW area percentage (%), and the relative amount of each polysaccharide was obtained.

#### 3.4.5. Fourier-Transform Infrared Spectroscopy Analysis

The crude EPS produced by *T. borchii* at the sucrose and yeast extract concentration of 80 and 20 g/L, respectively, extracted with NaOH and collected on the 28th day of cultivation by submerged fermentation, was subjected to FTIR (FT-720; Horiba, Ltd., Tokyo, Japan) for the analysis of the structure and different functional groups in the EPS. The sample and potassium bromide (KBr) were mixed in the ratio 1:100, ground into powder by agate mortar, and then pressed into a pellet. The FTIR analysis conditions set were as follows: scanning 32 times per measurement; resolution set at 2 cm^−1^; and scanning wavelength range 500~4000 cm^−1^.

### 3.5. Statistical Analysis

All the experiments were conducted in triplicates. Several flasks were run simultaneously, and daily sampling was done in triplicates by taking three flasks every time. A mean ± standard deviation was used to express each result. The statistical significance was assessed by the multiple comparisons Tukey post-hoc analysis of variance (ANOVA) using Origin Pro ver.9.0 (Origin Lab Corporation, Northampton, MA, USA). The differences of the results were considered statistically significant at *p*-values < 0.05.

## 4. Conclusions

The current study presents the first ever report on the cultivation of *T. borchii* by submerged fermentation for the production of polysaccharides of biological importance. Various carbon and nitrogen sources were screened and their concentration was optimized in order to achieve the maximum amount of mycelial biomass and polysaccharides (EPS and IPS). The most suitable carbon and nitrogen sources were found to be sucrose (80 g/L) and yeast extract (20 g/L), respectively, which yielded maximum mycelial growth and enhanced EPS and IPS production. The molecular weight of the EPS analyzed by the GPC method revealed that with the increase in concentration of yeast extract to 20 g/L, the molecular weight distribution shifted to the high-molecular-weight range. The highest average molecular weight of the EPS was 92 kDa, which was achieved in the EPS extracted using NaOH at a yeast extract concentration of 20 g/L on the 28th day of submerged fermentation. The results of this study demonstrated the production of high-molecular-weight EPS, which was found to be β-(1–3)-glucan after structural characterization. Moreover, the EPS (β-(1–3)-glucan) produced in this study can be screened for various functional properties, such as anti-cancer, antioxidant, immunomodulation, and glucose lowering, to determine its potential role in the biomedical field.

## Figures and Tables

**Figure 1 ijms-24-04875-f001:**
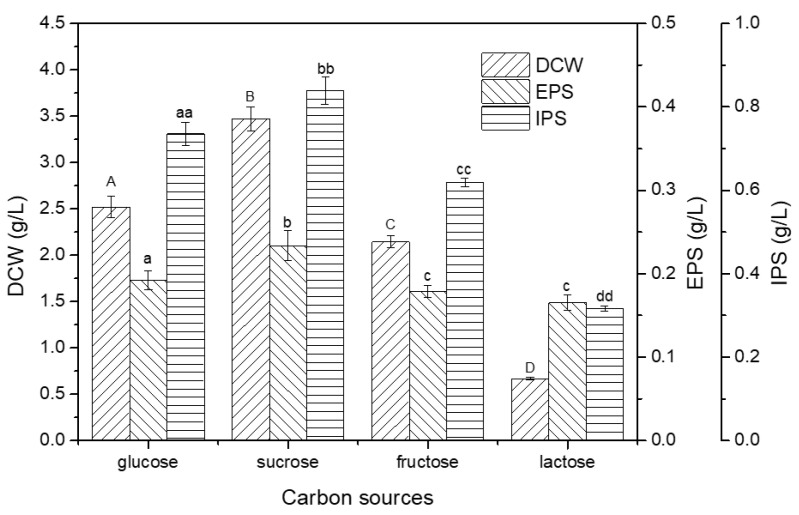
Effect of different carbon sources on mycelial growth (DCW: dry cell weight) and polysaccharides (EPS: exopolysaccharide; IPS: intracellular polysaccharide) production. Different letters above bars in the upper case indicate significant difference (*p* < 0.05) in DCW, different letters above bars in lower case indicate significant difference (*p* < 0.05) in EPS, and different letters above bars in double lower case indicate significant difference (*p* < 0.05) in IPS.

**Figure 2 ijms-24-04875-f002:**
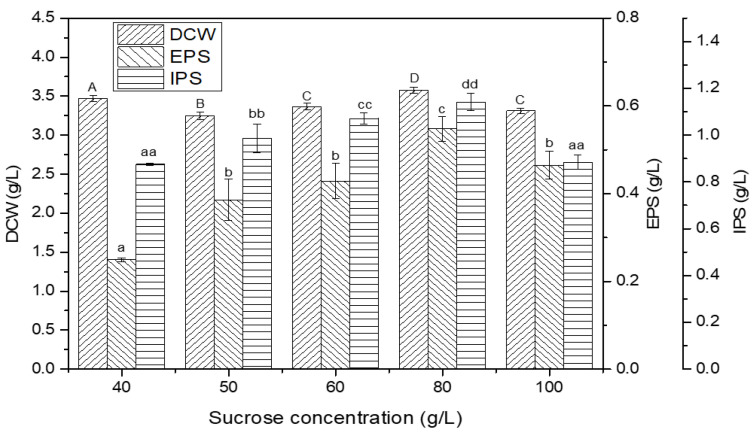
Effect of different sucrose concentrations on mycelial growth (DCW: dry cell weight) and polysaccharides (EPS: exopolysaccharide; IPS: intracellular polysaccharide) production. The letters above bars are the same indication as shown in Figure 1.

**Figure 3 ijms-24-04875-f003:**
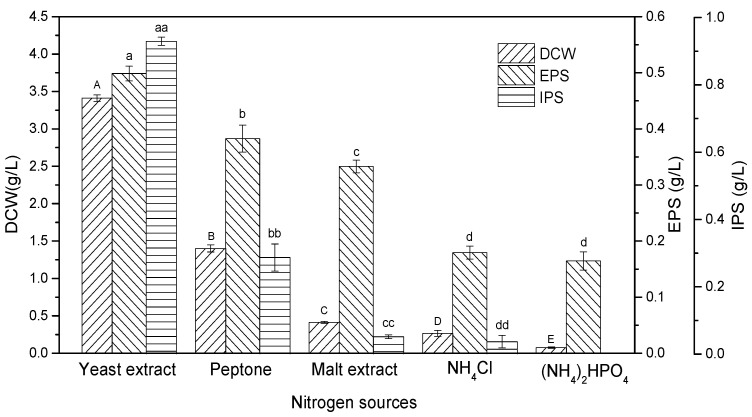
Effect of different nitrogen sources on mycelial growth (DCW: dry cell weight) and polysaccharides (EPS: exopolysaccharide; IPS: intracellular polysaccharide) production. The letters above bars are the same indication as shown in Figure 1.

**Figure 4 ijms-24-04875-f004:**
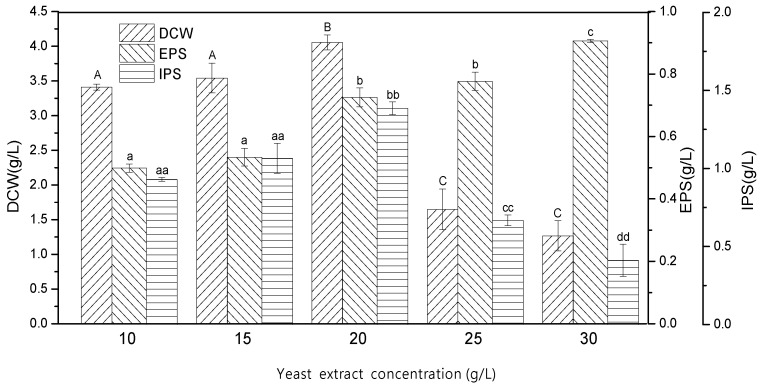
Effect of different yeast extract concentrations on mycelial growth (DCW: dry cell weight) and polysaccharides (EPS: exopolysaccharide; IPS: intracellular polysaccharide) production. The letters above bars are the same indication as shown in Figure 1.

**Figure 5 ijms-24-04875-f005:**
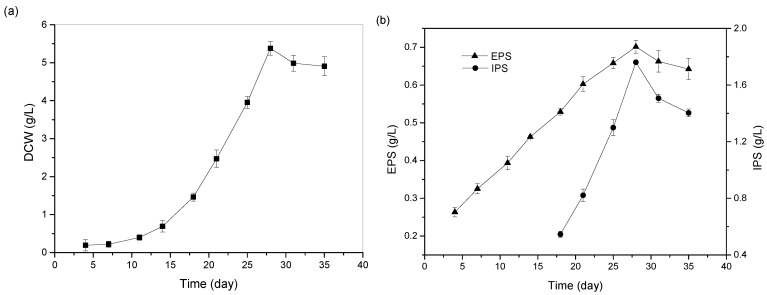
Time profile of *T. borchii* (**a**) enhanced dry cell weight; (**b**) polysaccharides production under submerged fermentation.

**Figure 6 ijms-24-04875-f006:**
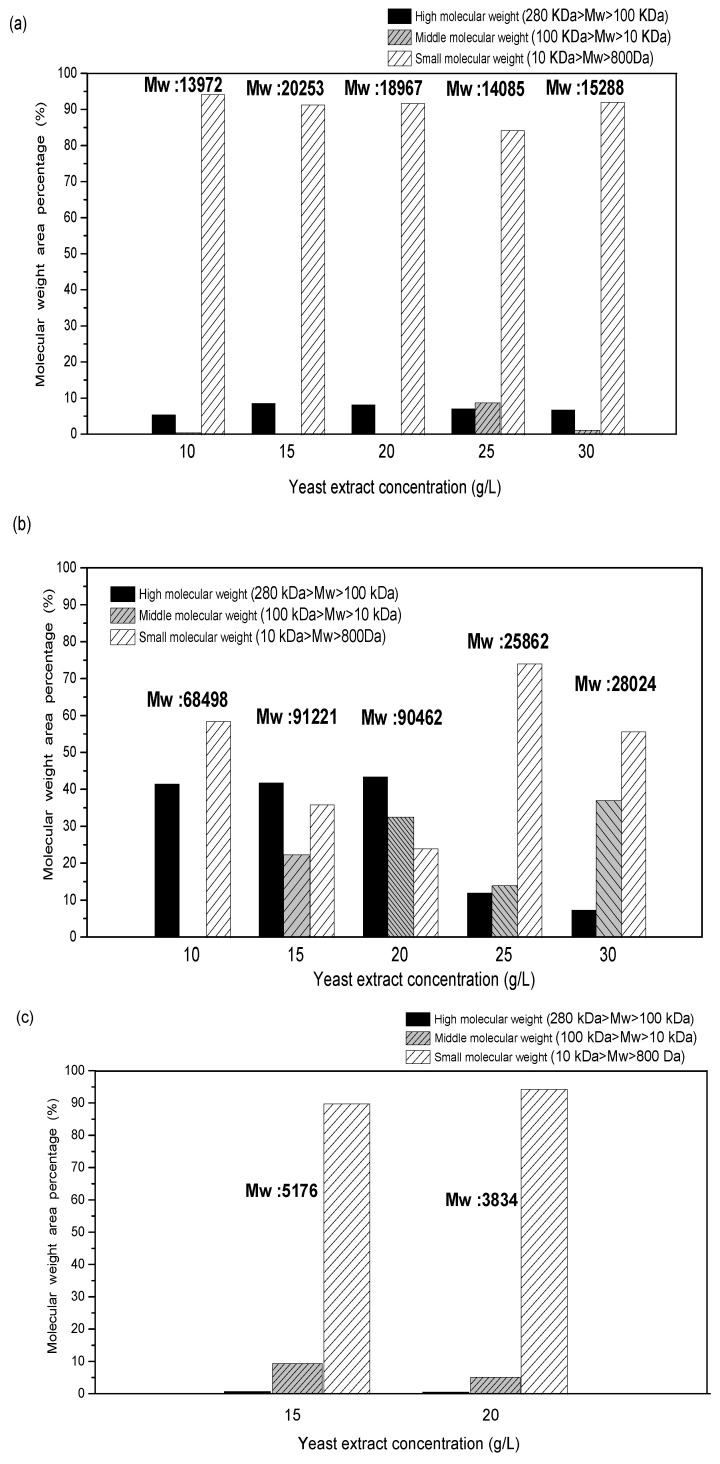
Molecular weight distribution of (**a**) water-extracted, (**b**) NaOH-extracted, and (**c**) HCl-extracted EPS produced at different concentrations of yeast extract.

**Figure 7 ijms-24-04875-f007:**
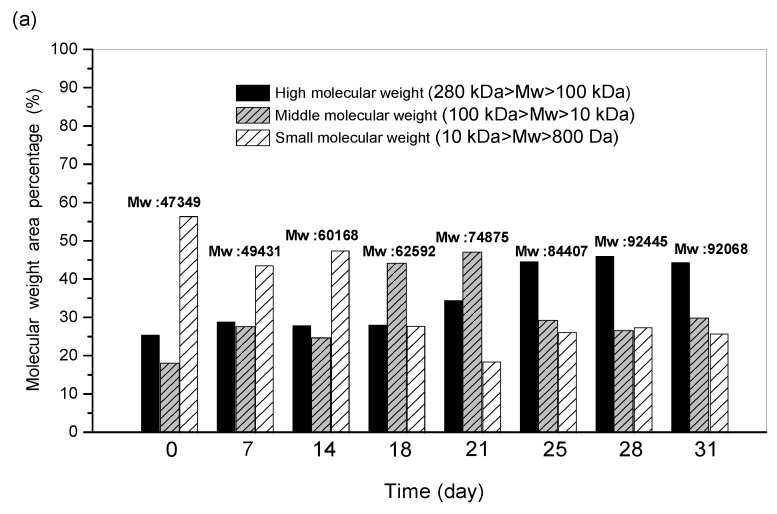
(**a**) Average molecular weight of EPS produced; (**b**) Molecular weight distribution of EPS produced during different days of submerged fermentation.

**Figure 8 ijms-24-04875-f008:**
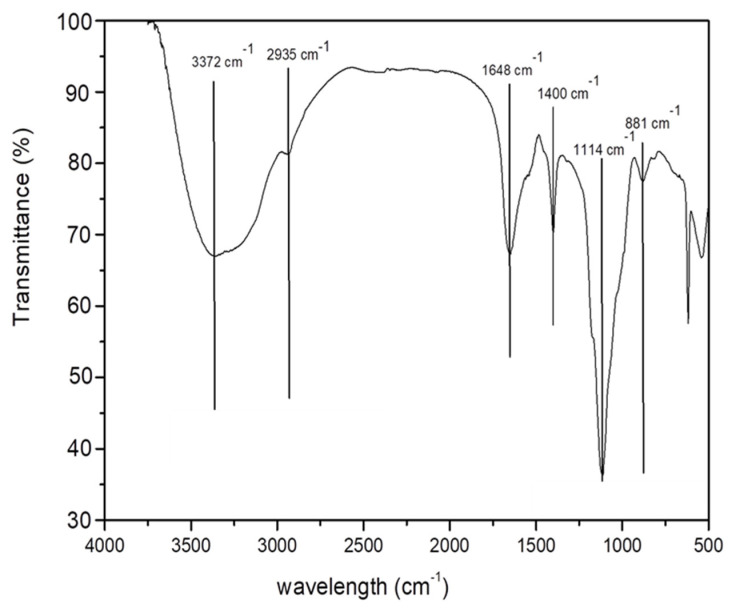
FTIR analysis of exopolysaccharides (EPSs) produced from *T. borchii* cultivated by submerged fermentation.

**Table 1 ijms-24-04875-t001:** Effect of different carbon sources on the total yield of mycelial biomass and polysaccharides production from *T. borchii* submerged cultivation.

Carbon Source	Yx/s (%)	Y_EPS_/s (%)	Y_EPS_/x (g/g)
glucose	6.30	0.48	0.076
sucrose	8.68	0.59	0.068
fructose	5.36	0.45	0.084
lactose	1.67	0.41	0.245

Yx/s: biomass yield based on substrate, Y_EPS_/s: EPS yield based on substrate, Y_EPS_/x: EPS yield based on biomass.

**Table 2 ijms-24-04875-t002:** Effect of different C/N ratios on mycelial biomass and polysaccharides production when sucrose and yeast extract were used for *T. borchii* submerged cultivation.

C/N Ratio (*w*/*w*)	Dry Cell Weight (g/L)	EPS (g/L)	IPS (g/L)
Case I			
4 (40:10)	3.47 ± 0.04	0.249 ± 0.41	0.875 ± 0.51
5 (50/10)	3.25 ± 0.05	0.386 ± 4.70	0.987 ± 6.13
6 (60/10)	3.37 ± 0.04	0.429 ± 3.95	1.071 ± 2.48
8 (80/10)	3.58 ± 0.04	0.547 ± 2.83	1.142 ± 3.62
10 (100/10)	3.31 ± 0.03	0.464 ± 3.15	0.884 ± 2.98
Case II			
8 (80/10)	3.41 ± 0.04	0.498 ± 13.24	0.929 ± 12.36
5.3 (80/15)	3.54 ± 0.21	0.603 ± 28.14	1.064 ± 96
4 (80/20)	4.06 ± 0.11	0.725 ± 30.17	1.385 ± 41.03
3.2 (80/25)	1.65 ± 0.29	0.776 ± 29.72	0.667 ± 34.38
2.66 (80/30)	1.27 ± 0.22	0.905 ± 4.85	0.411 ± 103.05

## Data Availability

Data are contained within the article.

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
