# Peer review of "High-Molecular-Weight Exopolysaccharides Production from Tuber borchii Cultivated by Submerged Fermentation"

_ijms, 2023, doi:10.3390/ijms24054875_

Round 1

Reviewer 1 Report

Dear authors,

thank you very much for these interesting manuscript about the exopolysaccharide production by Tuber borchii. I really enjoyed to read the results. However, the manuscript needs English improvement. I made a lot of suggestions and would be happy to read the manuscript again after major revision!!

Author Response

Reviewer 1

Dear authors,

Thank you very much for these interesting manuscript about the exopolysaccharide production by Tuber borchii. I really enjoyed to read the results. However, the manuscript needs English improvement. I made a lot of suggestions and would be happy to read the manuscript again after major revision!!

Response: Thank you so much for your valuable comments. We have made the changes in the revised manuscript as suggested.

My comments are:

Comment: in the title: Cultivation of Tuber borchii by submerged fermentation for high molecular weight exopolysaccharides the production

Response: We have updated the title of the study in the revised manuscript as per your suggestion.

Comment: In lines 14ff “Truffles are known worldwide for their peculiar taste, aroma and nutritious properties, which increase their economic value.”

Response: We have updated the statement in the revised manuscript as per your suggestion.

Comment: In lines 17ff “Therefore, in the current study the cultivation of Tuber borchii in submerged fermentation was executed to enhance the production of mycelial biomass …”

Response: We have updated the statement in the revised manuscript as per your suggestion.

Comment: In lines 24ff “Molecular weight analysis performed by gel permeation chromatography method revealed high proportion of high molecular weight EPS, if 20 g/L yeast extract was used as media and NaOH extraction step was carried out.”

Response: We have updated the statement in the revised manuscript as per your suggestion. However, in place of ‘if’ web have added ‘when’ after consulting with expert in English language.

Comment: In line 27 add a comma after glucan

Response: As suggested we have added comma after glucan in the revised manuscript.

Comment: In line 29 “To the best of our knowledge, this study represents the first FTIR analysis for the structural characterization ….”

Response: We have updated the statement as per your suggestion in the revised manuscript.

Comment: In line 34ff you normally describe hierarchies from top to the bottom not from bottom to the top. Moreover the phylum is Ascomycota und is not written in italic. So “Truffles are ectomycorrhizal fungi that belong to phylum Ascomycota, order Pezizales, family Tuberaceae genus Tuber [1,2].”

Response: We have updated the statement as per your suggestion in the revised manuscript.

Comment: In line 35 “Truffles live in symbiotic relationships with …” or “Truffles form symbiotic relationships with …”

Response: As per your suggestion, we have updated the statement in the revised manuscript.

Comment: In lines 37f “True truffles are those that belong to the genus Tuber, while other fruiting bodies such as from the genus Melanogaster are called false truffles, truffle-like

Response: As per your suggestion, we have updated the statement in the revised manuscript.

Comment: In line 39 replace “owing” with “because” or “due to”

Response: As suggested, we have replaced “owing” with “due” in the revised manuscript.

Comment: In line 43 laccases are not hydrolase!

Response: We have updated the statement in the revised manuscript.

Comment: Some doubling is found so in the lines 40f “They are a great source of vitamins, minerals, amino acids, and polysaccharides, and thus, possess a promising potential for application in pharmacology and medicine. “ and lines 44ff , moreover the sentences with the enzymes in between should either go to another place in the manuscript or should be removed, because it breaks the flow of reading why truffles are so important

Response: We have removed the doubling of lines and mentioned all the essential properties of truffle together. Moreover, we have also shifted the statement related to enzymes to another place and kept after Truffle properties in the revised manuscript.

Comment: In line 47 I suggest to replace “chemicals” with “componds”, as chemicals are normally high pure

Response: As suggested we have replaced “chemicals” with “compounds” in the revised manuscript.

Comment: In line 50 I suggest to write “appreciated” instead of “regarded”

Response: As suggested we have replaced “appreciated” with “regarded” in the revised manuscript.

Comment: In lines 50f “It is a whitish truffle that occurs …” or “It is a whitish truffle that can be found …”

Response: We have updated the statement in the revised manuscript as suggested.

Comment: In line 59 “reduce the overexploitation” instead of “lessen the overuse”

Response: We have updated the statement in the revised manuscript as suggested.

Comment: Inline 62 “can help overcome the limitations” instead of “aid in ruling out the limitations”

Response: As suggested, we have updated the statement in the revised manuscript.

Comment: In line 71ff use a protected hyphen between β and glucan

Response: We have used a protected hyphen between β and glucan as suggested.

Comment: In line 72ff “mixed α-, β-glucan”“(1→3)/(1→6)-β-D-glucans” and the D should be in a small capital D

Response: The changes have been made as per your suggestion in the revised manuscript.

Comment: In line 74 add for what NK is the abbreviation

Response: We have now written full form of NK cells in the revised manuscript.

Comment: In line 76f “Heteroglycans, which consist of sugars such as mannose, fructose, galactose and glucose, are also present in the truffle.”

Response: As suggested, we have updated the statement in the revised manuscript.

Comment: In line 84 “… activity, whereas …“

Response: As suggested, we have made the change in the revised manuscript.

Comment: In line 87ff “… polysaccharide, T. borchii was cultivated in submerged fermentation in the current study.”

Response: As suggested, we have made the change in the revised manuscript.

Comment: In line 88 “The effects of” instead of “The impacts of”

Response: As suggested, we have made the change in the revised manuscript.

Comment: In line 100 replace “ascertain” with “determine“

Response: We have replaced “ascertain” with “determine” in the revised manuscript.

Comment: In line 102 I suggest to write “It was found that the DCW of truffle with sucrose as carbon source, with 3.47 g/L, was significantly higher than the other three carbon sources after 28 days of submerged fermentation (Figure 1).” moreover in the Figure 1 and the materials and methods, I could not find any real statistical analysis, please add how you have done this and how much would be the difference in significance. The same is true for in line 153 and figure 2 “is significantly”. In the paragraph 3.5. Statistical analysis you only explain the how you calculated the mean and the standard deviation

Response: We have updated the statement in the revised manuscript. All the experiments were performed in triplicate; data are expressed as means ± SD (standard deviation). The statistical significance was assessed by the multiple comparisons Tukey post-hoc analysis of variance (ANOVA) using OriginPro ver.9.0 (OriginLab Corporation, Northampton, MA, USA). Differences of the results were considered statistically significant at p-values < 0.05. Different superscripts in the upper case indicate significant difference (p < 0.05) in DCW, different superscripts in lower case indicate significant difference (p < 0.05) in specific yield of EPS and different superscripts in double lower case indicate significant difference (p < 0.05) in specific yield of IPS. We have now included the statistical analysis in figures and material and method section as per your suggestion.

Comment: In lines 108f use a protected hyphen between β and galactosidase

Response: We have used a protected hyphen between β and galactosidase as suggested.

Comment: I check that a genome of T. borchii is publically available and there you can find at least 3 GH2 (to which the some β-galactosidase belong and at least one [Tubbor11127072] is annotated as), so do you really think that T. borchii has no β-galactosidase?

Response: According to this comment, the sentence of “absence of β-galactosidase activity in truffle” has now been revised to “low β-galactosidase activity expression in truffle”

Comment: In line 113f “The consumption of sucrose as a carbon source resulted in a maximum EPS production that was about 1.41 times higher than the lowest yield, with lactose as a media component (165.15 9 mg/L).”

Response: We have updated the statement in the revised manuscript as per your suggestion.

Comment: In line 117 “(838.78 ± 32.3 mg/L),”, as for the other quantities you have always spaces between the determined value and the error

Response: We have now added the space between the determined value and the error in this result.

Comment: In line 119 “ … (316.73 ± 6.2 mg/L), which …„

Response: We have updated the statement in the revised manuscript as per your suggestion.

Comment: In line 123 a linkage between the two sentences is missing, moreover what is “viz.”?

Response: The statement has been now updated and linked with the previous statement in the revised manuscript.

Comment: In line 123ff, you hypothesis is that the “choice and consumption of carbon sources vary among different truffle varieties”, but Pleurotus sp. is not a Truffle. Moreover, I wanted to mention, that the culture conditions (temperature, shaking speed, light/dark) can have an influence on growth and metabolism, are the same in the three studies?

Response: As per your observation, we have mentioned about Pleurotus sp. which is not a Truffle as there are rare reports available on the use of different carbon/nitrogen sources for mycelial biomass and polysaccharide production from truffle. Therefore, in order to do a comparison with other edible fungi the said reference was cited. Additionally, we have now corrected our hypothesis in the revised manuscript. The sentence “different truffle varieties” was revised to “different mushroom varieties”.

Moreover, the culture conditions such as (temperature, shaking speed, light/dark) indeed have effects on the growth and metabolism of the Truffle species. However, only the shaking speed used for the growth of Truffle in the current study was different from the T. sinense reference (https://doi.org/10.1016/j.procbio.2008.01.021 ), but the temperature for growth was same (25 ºC).

Comment: In lines 135,167,205 and 233 in the figure caption you should explain DCW, EPS and IPS

Response: The changes have been made as per your suggestion in the revised manuscript.

Comment: In lines 137f use a protected space between 100 and g/L

Response: We have used a protected space between 100 and g/L as suggested.

Comment: In line 141 “difference” instead of “different”

Response: The word has been now updated in the revised manuscript.

Comment: In lines 143f use a protected space between 28.34 and mg/L

Response: We have used a protected space between 28.34 and mg/L as suggested.

Comment: In line 148 use a space between 40 and g/L

Response: We have added space between 40 and g/L as suggested.

Comment: In line 154 use a space between 100 and g/L

Response: We have added space between 100 and g/L as suggested.

Comment: In line 156 „ … source; which …”

Response: The change has been made in the revised manuscript.

Comment: In line 158 „ reduce” instead of render

Response: The word has been now updated in the revised manuscript.

Comment: In line 159 “The results were in line with the …” and “in submerged fermentation”

Response: The words have been now updated in the revised manuscript.

Comment: In line 170 “effects” instead of impacts

Response: The word has been now updated in the revised manuscript.

Comment: In line 182 „… achieved, which was about…”

Response: The change has been made in the revised manuscript.

Comment: In line 185 „obtained” instead of “attained”

Response: The word has been now updated in the revised manuscript.

Comment: In line 189 “… source, no IPS was detectable due to …”

Response: The word has been now updated in the revised manuscript.

Comment: In lines 190f “… may have promoted cell growth and polysaccharide formation by supporting the synthesis of essential amino acids since cell density and polysaccharide production were low in the presence of organic nitrogen.”

Response: The statement has been now updated in the revised manuscript.

Comment: In lines 203/223/248 in submerged

Response: The word has been now updated in the revised manuscript.

Comment: In lines 216f use a protected space between 41.03 and mg/L

Response: We have added protected space between 41.03 and mg/L as suggested.

Comment: Figure 5, Figure 6 and Figure 7 should be merged and labeled with a and b (and c for 6) with only one caption, respectively, moreover only 7b is in different colors, please make it in line with the others

Response: All the figures have been updated as per your suggestion in the revised manuscript.

Comment: In line 274 viz.?

Response: The word has been now updated in the revised manuscript.

Comment: In lines 216f use a protected space between 100 and KDa

Response: We have added protected space between 100 and KDa as suggested.

Comment: In lines 291/294 no space between the digits and the %

Response: The change has been made in the revised manuscript.

Comment: In line 314 “It was shown from the results of the present study …”

Response: The statement has been now updated in the revised manuscript.

Comment: In line 353 I suggest to write a molecule with several free hydroxyl groups instead of “polyhydroxilic molecule”

Response: As per your suggestion, the change has been made in the revised manuscript.

Comment: In line 354 “2935 cm-1, which …“

Response: The change has been made in the revised manuscript.

Comment: In lines 359f use a protected hyphen between β and configuration

Response: We have used a protected hyphen between β and configuration as suggested.

Comment: In line 354 “… region, which confirmed that the EPS …”

Response: The change has been made in the revised manuscript.

Comment: In line 367 T. borchii

Response: The change has been made in the revised manuscript.

Comment: In line 385/391 B1

Response: The change has been made in the revised manuscript.

Comment: In line 385 for Watson Life Science Corporation and Sigma-Aldrich please add the country and city

Response: It has been revised to “Becton, Dicknson and Company, Sparks, MD, USA and Merck Ltd., Taipei, Taiwan”

Comment: In 386 please explain more/list what “local dealers”

Response: The statement has been removed from the revised manuscript.

Comment: In lines 389f use a protected space between 100 and mL

Response: We have used a protected space between 100 and mL as suggested.

Comment: In lines 407 “…(30g/L), which …”

Response: The change has been made in the revised manuscript.

Comment: In lines 420f use a protected space between 20 and min

Response: We have used a protected space between 100 and mL as suggested.

Comment: In line 424 explain DI

Response: Here, DI stands for deionized water, which has now been explained in the revised manuscript.

Comment: In lines 432ff, did you not precipitate the IPS, after maceration of the mycelium?

Response: We have done the precipitation process in the treatment. The sentence was added in the text as follows. “The supernatant was collected and subjected to the same ethanol precipitation and washing process to obtain the crude IPS as described in section 3.4.2.”

Comment: In line 439 “were connected in a serie” not jointed

Response: The change has been made in the revised manuscript.

Comment: In line 450 T. borchii instead of truffle

Response: The change has been made in the revised manuscript.

Comment: In line 451 (name of the machine and manufacturer)!!

Response: The machine name and manufacturer name for FTIR has been added in the revised manuscript.

Comment: In line 452 KBr?

Response: KBr is potassium bromide, which has now been mentioned in the revised manuscript.

Comment: In line 461 T. borchii instead of Tuber borchii

Response: The change has been made in the revised manuscript.

Comment: In line 462 in submerged

Response: The change has been made in the revised manuscript.

Comment: In line 470 “…was 92 KDa, which …”

Response: The change has been made in the revised manuscript.

Comment: In line 471ff “The results of this study demonstrate the production of high molecular weight EPS, which was found to be β-(1-3)-glucan after structural characterization.”

Response: The statement has been updated in the revised manuscript.

Comment: In line 473ff ”Moreover, the EPS (β-(1-3)-glucan) produced in this study can be screened for various functional properties such as anti-cancer, anti-oxidant, immunomodulation and glucose lowering to determine its potential in the biomedical field.”

Response: The statement has been updated in the revised manuscript.

Comment: I could not download the Supplementary Materials, did you upload them?

Response: Supplementary Material file is uploaded as a separate word file and contains Table S1 and strain identification report.

Comment: For me Table 1 is redundant with Figure 4, I suggest shifting the Table 1 in the SI

Response: Table 1 is now added in supplementary material.

Reviewer 2 Report

The manuscript entitled "Cultivation of Tuber borchii under submerged fermentation for high molecular weight exopolysaccharide production" is appropriate for the journal. There is novelty and originality in the research developed by the researchers. However, there are some observations that need to be reviewed and addressed. The main question is the lack of use of statistical tools to strengthen their observations and support their conclusions. It is crucial to incorporate this element into the manuscript.

Other comments can be consulted in the body of the manuscript.

Author Response

Reviewer 2

Comment: The manuscript entitled "Cultivation of Tuber borchii under submerged fermentation for high molecular weight exopolysaccharide production" is appropriate for the journal. There is novelty and originality in the research developed by the researchers. However, there are some observations that need to be reviewed and addressed. The main question is the lack of use of statistical tools to strengthen their observations and support their conclusions. It is crucial to incorporate this element into the manuscript.

Other comments can be consulted in the body of the manuscript.

See PDF file

Response: Thank you so much for reviewing our manuscript and providing valuable suggestions. Each comment raised by you in the PDF file has been addressed.

Reviewer 3 Report

Dear authors, 

you have presented great work and I am congratulating to you, although some parts could be shortened like 2.6.1. and 2.6.2. since main points should be underlined, the others can be seen from Figures. 

Why did you choose T. borchii, please explain at the end of the Introduction section.

Author Response

Reviewer 3

Dear authors, 

Comment: you have presented great work and I am congratulating to you, although some parts could be shortened like 2.6.1. and 2.6.2. since main points should be underlined, the others can be seen from Figures. 

Response: Thank you so much for your valuable feedback. We have taken care of your suggestion and shortened sections 2.6.1 and 2.6.2 as per requirement.

Comment: Why did you choose T. borchii, please explain at the end of the Introduction section.

Response: We have now added a paragraph where we have introduced Truffles in the Introduction section indicating the importance of T. borchii and why we chose it for the current study in the revised manuscript as per your suggestion.

Reviewer 4 Report

The title does not reflect the content of the manuscript, since the objective was the production of biomass and polysaccharides, both extra- and intracellular, in addition, in general, the highest content of EPS was those with low MW.

Line 14 indicates the EPS/IPS ratio, however, the values obtained are not shown anywhere in the manuscript, in the same way, line 20 mentions the C/N ratio, however, it is ignored in the experimental design.

It is suggested that all the results are reported in the same units, there are occasions when they are reported in g/L and others in mg/L (in the text and in the figures), which causes confusion.

Review the entire manuscript, there are grammatical errors, for example, in line 50 it should be "is" instead of "are", put the genus in italics on line 37.

The information in lines 98-102 are methodology, not results or discussion.

In line 103 the value in brackets is from DCM not from sucrose.

Explain with background the possibility that galactose can inhibit the growth of the truffle (lines 107-108); it is suggested to determine the activity of b-galactosidase.

In all the results, the authors speak of yields, however, what they report are values, so it is suggested to calculate the yields Yp/x (yield of EPS or IPS with respect to biomass), for example with EPS the Yp /x with sucrose is Yp/x= (233.62/3.47)=67.32; while for lactose it is Yp/x=(165.15/0.67)=235.9, which reflects the system with lactose being 3.5 times more productive. Calculate yields and explain.

An explanation must be given as to why high EPS values are obtained with so little biomass produced with lactose.

The content of residual sugars in the fermentations must be determined and thus calculate Yp/s (yield of EPS with respect to the consumed substrate) and Yx/s (yield of biomass with respect to the consumed substrate).

All the results must be statistically analyzed, they must be analyzed with ANOVA and with the Tukey or Duncan test.

It is essential that when increasing the concentration of the carbon source, the C/N ratio is maintained, since poor growth can also be due to a lack of nitrogen.

Review and correct the way of citing (lines 196 and 220).

There is a very limited discussion, it is suggested to improve it and analyze the results with that of other authors.

In line 400, are the units correct?

Author Response

Reviewer 4

Comment: The title does not reflect the content of the manuscript, since the objective was the production of biomass and polysaccharides, both extra- and intracellular, in addition, in general, the highest content of EPS was those with low MW.

Response: We agree with your comment as the current study reports the production of biomass and polysaccharides, both extra and intracellular, however, the main focus of the study was the production of EPS and screening of best carbon and nitrogen sources for enhanced EPS production. Therefore, the EPS produced from the truffle was also subjected to GPC and FTIR analysis for the estimation of molecular weight and functional groups, respectively. The reason behind these analyses for only EPS was just confirm if the EPS produced by the truffle is β-(1-3)-glucan which possesses various bio-medical characteristics and thus, can be further investigated for the same. We also agree that the highest content of EPS was those with low MW but this was in general for all yeast extract concentrations screened in the study. However, when 20 g/L yeast extract concentration was used and EPS was extracted using NaOH, the proportion of high molecular weight EPS was high as compared to small and middle molecular weight EPS (Figure 6b) which was the actual aim of the current study. Moreover, it has also been shown that on the 28th day of submerged fermentation the average MW of EPS was about 92 KDa. Nevertheless, we have modified the title to make it more clear and specific for the readers.

Comment: Line 14 indicates the EPS/IPS ratio, however, the values obtained are not shown anywhere in the manuscript, in the same way, line 20 mentions the C/N ratio, however, it is ignored in the experimental design.

Response: The word “EPS/IPS” used in the manuscript refers to “EPS and IPS” and not to their ratio as word “ratio” was not written next to “EPS/IPS” anywhere in the manuscript. “/” was just used as an alternative to word “and”. Similar was the case for C/N, nowhere in the manuscript “ratio” was mentioned next to “C/N”. However, in order to avoid this confusion, these words have been now corrected in the revised manuscript.

Comment: It is suggested that all the results are reported in the same units, there are occasions when they are reported in g/L and others in mg/L (in the text and in the figures), which causes confusion.

Response: All the results are now expressed in the same units in text and figures throughout the revised manuscript.

Comment: Review the entire manuscript, there are grammatical errors, for example, in line 50 it should be "is" instead of "are", put the genus in italics on line 37.

Response: The corrections have been made in the revised manuscript. Moreover, the manuscript has been thoroughly reviewed and corrected for any grammatical errors.

Comment: The information in lines 98-102 are methodology, not results or discussion.

Response: We have modified the lines by giving a little background regarding what the result is all about in the revised manuscript.

Comment: In line 103 the value in brackets is from DCM not from sucrose.

Response: We have corrected the line in the revised manuscript.

Comment: Explain with background the possibility that galactose can inhibit the growth of the truffle (lines 107-108); it is suggested to determine the activity of b-galactosidase.

Response: We have cited a study which has reported galactose to be unsuitable for the growth of medicinal mushroom Phellinus. We have also updated the statement in the manuscript as “The poor mycelial growth observed in presence of lactose could be possibly either due to the inability of β-galactose (one of the monomers of lactose) to favour the mycelial growth or low β-galactosidase activity expression in truffle Tuber brochii to hydrolyze lactose into glucose and galactose before it can be used effectively as a carbon source [23]. A study reported galactose along with lactose and maltose to be unsuitable for the growth of a medicinal mushroom of genus Phellinus as they had less impact on promoting mycelial growth”.

Moreover, we had only observed the slow biomass growth effect from the results via the use of lactose and the reasons were adopted from the previous reports. To the real situation, the activity of β -galactosidase should be tested. However, the low consumption of lactose was not our main concern in the text, we just tried to investigate lactose effect and the reason might be left for our extensive study.

Comment: In all the results, the authors speak of yields, however, what they report are values, so it is suggested to calculate the yields Yp/x (yield of EPS or IPS with respect to biomass), for example with EPS the Yp /x with sucrose is Yp/x= (233.62/3.47)=67.32; while for lactose it is Yp/x=(165.15/0.67)=235.9, which reflects the system with lactose being 3.5 times more productive. Calculate yields and explain.

Response: The yield in the manuscript was used in terms of the EPS or IPS production by Truffle. The statement has been corrected and word “yield” has been replaced with a more suitable word in the revised manuscript. We have explained the results in terms of amount of biomass, EPS and IPS and not in terms of yield.

Comment: An explanation must be given as to why high EPS values are obtained with so little biomass produced with lactose.

Response: Even though the lactose was not suitable for the growth of the biomass which could be possibly due to the inability of galactose to favour biomass growth (as cited ref.), however, the biomass could still use the other hydrolyzed sugar, such as glucose, to produce EPS. Thanks to the reviewer suggestion, this provides us a good direction to further explore the lactose effect in T. brochii cultivation.

Comment: The content of residual sugars in the fermentations must be determined and thus calculate Yp/s (yield of EPS with respect to the consumed substrate) and Yx/s (yield of biomass with respect to the consumed substrate).

Response: We don’t have the residual sugars level at hand, however, we can predict the yield of Yp/s and Yx/s according to the total sugars we added in (i.e., 80 g/L sucrose), which were Yp/s = 0.97% and Yx/s = 2%, respectively. These data indicate a very low yield for both biomass and products. However, this study was set to design an approach to obtain high molecular EPS for future medical or cosmetic applications, the low production yield might be our future concerns.

Comment: All the results must be statistically analyzed, they must be analyzed with ANOVA and with the Tukey or Duncan test.

Response: All the experiments were performed in triplicate; data are expressed as means ± SD (standard deviation). The statistical significance was assessed by the multiple comparisons Tukey post-hoc analysis of variance (ANOVA) using OriginPro ver.9.0 (OriginLab Corporation, Northampton, MA, USA). Differences of the results were considered statistically significant at p-values < 0.05. Different superscripts in the upper case indicate significant difference (p < 0.05) in DCW, different superscripts in lower case indicate significant difference (p < 0.05) in specific yield of EPS and different superscripts in double lower case indicate significant difference (p < 0.05) in specific yield of IPS. We have now included the statistical analysis in figures and material and method section as per your suggestion.

Comment: It is essential that when increasing the concentration of the carbon source, the C/N ratio is maintained, since poor growth can also be due to a lack of nitrogen.

Response: It is indeed essential to maintain the C/N ratio for growth. However, in this study when the concentration of the selected carbon source (sucrose) was increased, the nitrogen source concentration was kept constant same as per the basal medium for submerged fermentation in order to identify the optimal concentration of the carbon source. Same process was followed for the identification of the optimal concentration of the nitrogen source and the selected carbon source concentration was kept constant. In fact, in the cultivations, the media we designed were rich both in carbon and nitrogen sources. The lack of nutrient sources might only occur at late cultivation time. Therefore, the poor growth due to lack of nitrogen might only happen at the late growth phase. To this concern, we will try to observe the time profile for various C/N ratio effect at our further study.

Comment: Review and correct the way of citing (lines 196 and 220).

Response: The citation has been corrected in the revised manuscript.

Comment: There is a very limited discussion, it is suggested to improve it and analyze the results with that of other authors.

Response: We have added few more references to improve the present MS and analyzed their results. However, since we have attempted to explore the EPS production ability of Tuber borchii, in addition to characterizing the EPS using physicochemical technique, i.e., FTIR for the first time, rare reports were available in the literature specifically on this species and topic. We have compared and discussed our results with those reported on edible fungi/mushroom.

Comment: In line 400, are the units correct?

Response: The units are corrected in the revised manuscript.

Thank you so much for your valuable suggestions. We have revised the manuscript as per your suggestions.

Round 2

Reviewer 1 Report

Dear authors,

 thank you again for these interesting manuscript about the exopolysaccharide production by Tuber borchii. I really enjoyed to read the results. I have only some minor comments now.

In line 88ff “(1→3)/(1→6)-β-D-glucans” the D is too small, same to the other D’s

In line 104 use a protected space between T. and borchii

In line 117 DCW should be explain, as it is meanted here the first time

In line 123 T. borchii

Figure 6 and 7 should be on one page each

In line 511 “… T. borchii and the strain …”

Author Response

Review 1

Dear authors,

Comment: thank you again for these interesting manuscript about the exopolysaccharide production by Tuber borchii. I really enjoyed to read the results. I have only some minor comments now.

Response: Thank you so much for the appreciation. We have revised the manuscript as per your suggestions.

Comment: In line 88ff “(1→3)/(1→6)-β-D-glucans” the D is too small, same to the other D’s

Response: We have corrected the “D” in the revised manuscript.

Comment: In line 104 use a protected space between T. and borchii

Response: We have added protected space between T. and borchii

Comment:  In line 117 DCW should be explain, as it is meanted here the first time

Response: We have explained DCW as suggested in the revised manuscript.

Comment:  In line 123 T. borchii

Response: We have made the change as suggested in the revised manuscript.

Comment:  Figure 6 and 7 should be on one page each

Response: We have made the change as suggested in the revised manuscript.

Comment:  In line 511 “… T. borchii and the strain …”

Response: As suggested, we have made the change in the revised manuscript.

Reviewer 4 Report

The authors have attended to most of the observations and suggestions, however, it is suggested:

1. Given the experimental design, it was not considered to keep the C/N ratio constant, but it is necessary to calculate and report the resulting C/N ratio in each bioprocess, which will serve to discuss the biomass and polysaccharide production results.

2. Although the word "yield" was changed, it is necessary to calculate the yields and explain the values, for example the Yp/x with lactose is 3.5 times higher than with sucrose.

Author Response

Review 4

Comment: The authors have attended to most of the observations and suggestions, however, it is suggested:

Response: Thank you so much for your valuable feedback and suggestions. We have incorporated your suggestions in the revised manuscript.

Comment: 1. Given the experimental design, it was not considered to keep the C/N ratio constant, but it is necessary to calculate and report the resulting C/N ratio in each bioprocess, which will serve to discuss the biomass and polysaccharide production results.

Response: We have added the results for C/N ratio as shown in Table 1 and the discussion is addressed in the revised manuscript.

Comment: 2. Although the word "yield" was changed, it is necessary to calculate the yields and explain the values, for example the Yp/x with lactose is 3.5 times higher than with sucrose.

Response: We have calculated the yield as shown in Table 2 and the discussion is addressed in the revised manuscript as per your suggestion.